

# The mitochondrial genomes of two walnut pests, *Gastrolina depressa depressa* and *G. depressa thoracica* (Coleoptera: Chrysomelidae), and phylogenetic analyses

Qiqi Wang[1,2,3] and Guanghui Tang[1]

[1] Key Laboratory of State Forestry Administration on Management of Western Forest Bio-Disaster, College of Forestry, Northwest A&F University, Yangling, Shaanxi, China
[2] Key Laboratory of Zoological Systematics and Evolution, Institute of Zoology, Chinese Academy of Sciences, Beijing, China
[3] University of Chinese Academy of Sciences, Beijing, China

## ABSTRACT

In this study, the mitochondrial genomes (mitogenomes) of two walnut leaf insect pests, *Gastrolina depressa depressa* and *G. depressa thoracica*, were sequenced by Sanger sequencing technology. The mitogenome of *G. depressa thoracica* was complete at 16,109 bp in length, while the mitogenome of *G. depressa depressa* (14,277 bp) was partial. The genomic analyses indicated that both mitogenomes have the typical gene content and arrangement. The formerly identified elements, 'TAGTA' between *trnSer*(*UCN*) and *nad2*, and 'ATGATAA' between *atp8* and *atp6*, were more conserved than that between *nad4L* and *nad4*, which was 'ATGTTAA' in Coleoptera excluding Polyphaga. Phylogenetic analyses of the 13 protein-coding genes from 36 coleopteran species well supported a close affinity between the subfamily Chrysomelinae including *G. depressa thoracica* and *G. depressa depressa* and Galerucinae, as well as a sister relationship of ((Eumolpinae + Cryptocephalinae) + Cassidinae) within Chrysomelidae.

Corresponding author
Guanghui Tang,
tanggh@nwsuaf.edu.cn

## INTRODUCTION

Both *Gastrolina depressa thoracica* and *G. depressa depressa* belong to Chrysomelinae (Coleoptera: Chrysomeloidea) and are the major insect pests of walnut in China. The larvae and adults feed on walnut leaves, and seriously influence walnut growth and yield. Compared with other species of Chrysomelinae in the boundary between Palaearctic and oriental regions, they have a typical flat back. Furthermore, *G. depressa thoracica* has a black prothorax, while *G. depressa depressa* has a yellow prothorax (*Chen, 1974*). *Ge et al. (2003)* comprehensively compared the morphological differences and the distributions of *G. depressa thoracica* and *G. depressa depressa*. Overall, in China *G. depressa depressa* are widely distributed in most areas south of the Yellow River, while *G. depressa thoracica* live

in north of the Yangtze River. They both have filamentous antennae, and similar structure of the hind wing, claw and upper lip, but there are large differences in the structure of lower lip (cilia: long and dense in *G. depressa thoracica*, short and sparse in *G. depressa depressa*) and lower jaw (between the anterior chin and lower lip: triangle in *G. depressa thoracica*, circular soliton in *G. depressa depressa*). However, there is no information about their mitochondrial genomes (mitogenomes) or any other molecular data to support their classification.

Mitogenomes have been widely used as molecular markers for phylogenetics and phylogeographics because of maternal inheritance, conserved gene order and orientation, low recombination rate and high mutation rate (*Avise, 1989*; *Crozier & Crozier, 1993*; *Jin et al., 2004*). In general, the mitogenomes range from 14 to 36 kb in length, encoding 13 protein-coding genes (PCGs), 22 transfer RNA genes (tRNAs), 2 ribosomal RNA genes (rRNAs) and a large non-coding region (control region or A+T-rich region) (*Wolstenholme, 1992*; *Boore, Lavrov & Brown, 1998*). Recent advances in sequencing technology have enriched insect genome and mitogenome datasets and accelerated insect molecular studies. Insect genome and mitogenome data are now widely used in species identification, population genetics and molecular evolution (*Ma et al., 2012*; *Timmermans, Lees & Simonsen, 2014*).

Coleoptera has as many as 380,000 described species and an estimated total number of about three million members. Cucujiformia is one of the most highly diversified infraorders of Polyphaga (*Crowson, 1960*; *ØDegaard, 2000*; *Bouchard et al., 2009*) and includes six superfamilies, among which the superfamilies Curculionoidea and Chrysomeloidea are the major plant-feeding beetles (*Grimaldi & Engel, 2005*). Several studies have reported the phylogenetic relationships within Chrysomeloidea by using morphological data (*Farrell & Sequeira, 2004*; *Gómez-Zurita et al., 2007*), 18S rRNA (*Hunt et al., 2007*), 28S rDNA (*Marvaldi et al., 2009*), and partial mitochondrial genes (*Bocak et al., 2014*; *Li et al., 2016*).

In this study, we sequenced the mitochondrial genomes of *G. depressa thoracica* and *G. depressa depressa* and clarified their mitogenome differences. The nucleotide and amino acid sequences of the 13 PCGs of two walnut pest insects were then aligned with data of 34 other coleopteran species for the phylogenetic analyses.

## MATERIALS AND METHODS

### Sample collection

Adults of *G. depressa thoracica* and *G. depressa depressa* were originally collected from the experimental station of Northwest A&F University, Shanyang County, Shaanxi Province, China (109°88′E, 33°53′N), and identified according to key morphological characteristics. All samples were stored in 100% ethanol at −20 °C.

### DNA extraction and PCR amplification

Total genomic DNA was extracted from an individual insect using a standard phenol-chloroform extraction (*Tamura & Aotsuka, 1988*). The specific primers of each mitochondrial gene (Table 1) were designed from the conserved regions after multiple alignments of coleopteran insect mitochondrial sequences. PCR was performed with a
**Table 1  Primers used in this study to sequence the mitochondrial genomes of *Gastrolina depressa thoracica* and *G. depressa depressa*.**

| Primer | Forward (5′ → 3′) | Reverse (5′ → 3′) | Tm(°C) | Length (kb) |
|---|---|---|---|---|
| *G. depressa thoracica* | | | | |
| 01 | GCCTGAAATGAAAGGATAATTTTGATA | GCTCGGGTATCTACATCTATTC | 55 | 2.2 |
| 02 | GTTAATATAAACTCTTAACCTTCAA | CCGCAAATCTCAGAGCATTG | 49 | 2.3 |
| 03 | ACAATTGGACATCAATGATACTG | ATGACCAGCAATTATATTAG | 51 | 1.1 |
| 04 | TTAGCACATTTAGTTCCACAAGG | TATAATTAGAGCATAATTTTGAAG | 50 | 1.9 |
| 05 | TTTAATTGAAACCAAATTAGAGG | TTTTTGTCGTAATGGTC | 50 | 4.1 |
| 06 | CGCTCAGGCTGATAGCCCCA | AATCGTACTCCGTTTGATTTTGC | 53 | 2.9 |
| 07 | CGAGGTAATGTACCCCGAACCCA | GTGCCAGCAGTTGCGGTTATAC | 58 | 2.8 |
| 08 | ACCTTTATAATTGAGGTATGAAC | ATAATAGGGTATCTAATCCTAG | 51 | 2.0 |
| *G. depressa depressa* | | | | |
| 01 | GCCTGATAAAAAGGATTATCTTGATA | TAAACTTCTGGGTGTCCAAAAAATCA | 52 | 2.0 |
| 02 | AATTGGGGGATTTGGAAATTG | CCACAAATTTCTGAACATTG | 49 | 2.0 |
| 03 | ACAATTGGACATCAATGATATTG | AGGGGCTTCTTTTTTCATAA | 47 | 2.3 |
| 04 | GCAGCTGCTTGATATTGACA | TTAGGATGGGATGGTTTGGG | 54 | 2.2 |
| 05 | TTTAATTGAAACCAAATAGAGG | GTTTGTGAGGGGGTTTTAGG | 55 | 3.4 |
| 06 | CCAGAAGAACAAATACCATG | TATCAATAGCAAATCCCCCCCA | 53 | 2.3 |
| 07 | TTCAGCAATATGAAATTTTGGATC | TTACCTTAGGGATAACAGCGTAA | 53 | 2.4 |
| 08 | CCGGTTTAAACTCAGATCATGTA | GTGCCAGCAGTTGCGGTTATAC | 57 | 1.8 |

3-step program: an initial denaturation at 94 °C for 2 min; followed by 35 cycles at 94 °C for 30 s, annealing temperature (Table 1) for 30 s, 72 °C for 1–4 min; and a final extension at 72 °C for 10 min. PCR products (1.1–4.1 Kb in length) were analyzed by 1% agarose gel electrophoresis, and sequenced.

## Genome assembly and annotation

All sequences were blasted in NCBI (https://www.ncbi.nlm.nih.gov/) and assembled using the program SeqMan in DNAStar package v7.1 (DNAStar Inc., Madison, WI, USA). The 13 PCGs were identified using ORF Finder (available on NCBI) and the rRNAs were determined by comparison with other coleopteran mitogenomes. The tRNAs and their cloverleaf secondary structures were predicted using tRNAscan-SE Search Online Server (*Lowe & Eddy, 1997*), while undefined tRNAs were further compared with tRNAs of other species, including *Atrijuglans hetaohei* (*Wang, Zhang & Tang, 2016*), *Dastarcus helophoroides* (*Zhang et al., 2015*) and *Anopheles minimus* (*Hua et al., 2016*). The secondary structure of all tRNAs were drawn using RNAviz v2.0 (*De Rijk, Wuyts & Wachter, 2003*). The composition of nucleotide sequences was described by skewness according to the following formulas (*Perna & Kocher, 1995*): AT-skew = $(A - T)/(A + T)$ and GC-skew = $(G - C)/(G + C)$. The A+T content and relative synonymous codon usage (RSCU) were calculated using MEGA v5.1 (*Tamura et al., 2011*).

## Phylogenetic analysis

The newly sequenced mitogenomes of *G. depressa thoracica* and *G. depressa depressa* were aligned with 32 mitochondrial genomes of Chrysomeloidea available in GenBank, with

*Tetraphalerus bruchi* (Coleoptera: Archostemata: Ommatidae) and *Abax parallelepipedus* (Coleoptera: Adephaga: Carabidae) as outgroups (Table S1). The nucleotide and amino acid sequences of 13 PCGs from all 36 species were aligned separately using ClustalW implemented in MEGA v5.1 (*Tamura et al., 2011*). Gblocks v0.91b (*Talavera & Castresana, 2007*) was used to refine the final alignments and identify the conserved sequences (or conserved motifs). After Gblocks analysis, the final matrix consisted of 7,289 nucleotides and 2,340 amino acids. The best-fit models (TVM + I + G for the nucleotide dataset and MtREV + I + G + F for amino acid dataset) were selected using MrModeltest v2.3 (*Nylander, 2004*) and ProtTest v2.4 (*Abascal, Zardoya & Plsada, 2005*) with specific parameters (Table S2).

Phylogenetic trees of PCGs were constructed using maximum likelihood (ML) and Bayesian inference (BI); these two methods (ML and BI) have different algorithms in phylogenetic analyses. In ML analyses, phylogenetic trees of nucleotide and amino acid sequences were constructed by PhyML v3.0 (*Guindon et al., 2010*) based on the best-fit model (as above) with 1,000 replicates. For BI analysis, MrBayes v3.2.6 (*Huelsenbeck & Ronquist, 2001*) was used to compute the probability distribution and get a sharper and stronger prediction; four simultaneous Markov chains ran for 10 million generations and sampled every 1,000 generations before discarding the first 25% ''burn-in'' trees. Node support was assessed by the value of Bayesian posterior probabilities (BPP). The consensus trees were viewed and edited by Figtree v1.4.3 (*Rambaut, 2009*).

# RESULTS AND DISCUSSION

## Gene content and nucleotide composition

The mitogenomes of *G. depressa thoracica* and *G. depressa depressa* were sequenced. The complete *G. depressa thoracica* mitogenome had 16, 109 bp, including 13 PCGs, 2 rRNAs and 22 tRNAs, and an A+T-rich region (Fig. 1, Table 2). In the mitogenome of *G. depressa thoracica*, there were four intergenic spacers (a total length of 23 bp), ranging from 1 bp to 17 bp, and the longest intergenic spacer was located between *trnSer* (*AGN*) and *nad1*. Furthermore, 15 pairs of genes overlapped each other, with a length ranging from 1 to 17 bp.

The sequenced mitogenome of *G. depressa depressa* was partial and 14,277 bp long, including 13 PCGs, 21 tRNAs, *rrnL* and partial *rrnS* (550 nucleotides from the 3′-end) (Fig. 1, Table 2). The A+T-rich region, *trnI* and partial *rrnS* (predicted to be about 200 nucleotides from the 5′-end) were failed to be sequenced although we have tried several pairs of species-specific primers. There were three intergenic spacers (23 bp), ranging from 1 bp to 17 bp, and the longest intergenic spacer was detected between *trnSer* (*AGN*) and *nad1*. Seventeen pairs of genes overlapped each other, with a length ranging from 1 to 20 bp.

Two 7-bp long overlaps (ATGATAA) were detected in both *G. depressa depressa* and *G. depressa thoracica*, which were also found in many other Polyphaga insects (Fig. 2). The overlaps were located between *atp8* and *atp6* on the H-strand and between *nad4L* and *nad4* on the L-strand, respectively. The overlapped sequences were thought to be translated as a bicstron (*Stewart & Beckenbach, 2005*). Another 5 bp long motif (TAGTA) was detected between *trnSer* (*UCN*) and *nad1* in mitogenomes of *G. depressa depressa* and

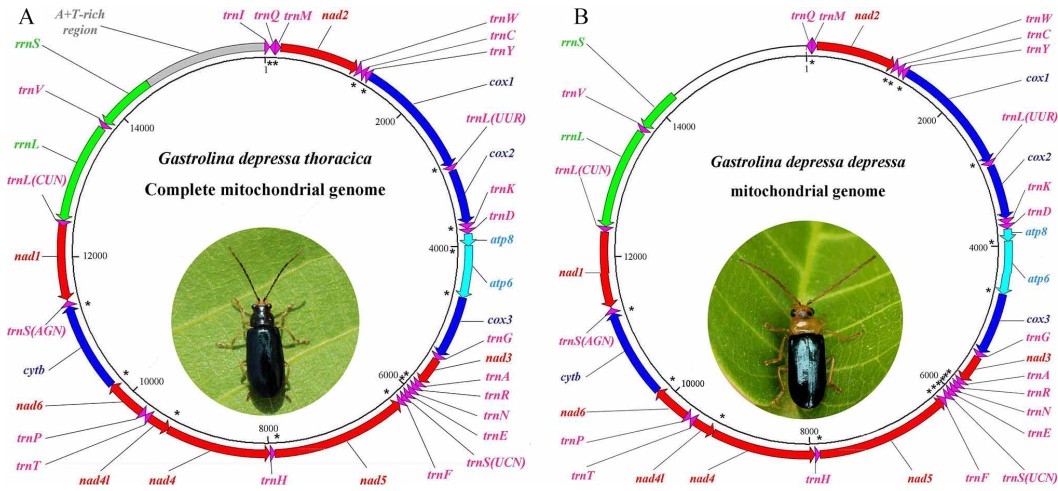

**Figure 1** Gene maps of the complete mitochondrial genome for **(A)** *G. depressa thoracica* and the incomplete mitochondrial genome for **(B)** *G. depressa depressa*. The PCGs and rRNAs are the standard abbreviations. Each tRNA is denoted as a one-letter symbol according to the IUPAC-IUB single-letter amino acid codes. Arrows indicate coding direction. The unmarked region, whole A+T-rich region, *trnI* and partial *rrnS*, are failed to be sequenced. Photos of the two insects were taken by Guanghui Tang.

*G. depressa thoracica*, which was also present in other coleopterans (Fig. 3). This consensus sequence has been proposed as the possible binding site of mtTERM because it is located at the end of the H-strand coding region in the circular mitogenome (*Taanman, 1999*). The sequenced motifs between *atp8* and *atp6*, and between *trnSer* (*UCN*) and *nad1* were relatively conserved in four suborders (41 Polyphaga, 24 Adephaga, two Archostemata and two Myxophaga) after mitogenomic comparisons. However, the motif 'ATGATAA' between *nad4* and *nad4L* was only found in the mitogenomes of Polyphaga insects, while 'ATGTTAA' was identified in insect mitogenomes from the other three suborders, Adephaga, Archostemata and Myxophaga (Fig. S1).

## Protein-coding genes

In the mitogenomes of *G. depressa thoracica* and *G. depressa depressa*, a total of 3,731 and 3,678 amino acids were encoded by 11,182 bp and 11,026 bp nucleotides, respectively. All genes had the same orientations and organization. The 13 PCGs ranged from 156 bp (*atp8*) to 1,722 bp (*nad5*) for *G. depressa thoracica* and from 156 bp (*atp8*) to 1,728 bp (*nad5*) for *G. depressa depressa* (Table 2). Except for *nad1* which used TTG as a start codon, all other PCGs had the typical ATN as the start codon. For example, *nad2*, *cox1*, *cox2* and *nad5* have 'ATT' as start codon, while the start codon 'ATA' was used for *nad3* and *nad6*, 'ATG' for *atp6*, *cox3*, *nad4*, *nad4L* and *cob*, and 'ATC' for *atp8* in *G. depressa thoracica* mitogenome. In *G. depressa depressa* mitogenome, start codon 'ATT' was used in six genes (*nad2*, *cox1*, *cox2*, *atp8*, *nad5* and *nad6*), 'ATG' for *atp6*, *cox3*, *nad4*, *nad4L* and *cob*, and 'ATC' for *nad3*. The stop codons for the two mitogenomes were TAA, TAG and incomplete termination codons, like TA or T (Table 3). The incomplete stop codons could be completed as TAA through post-transcriptional polyadenylation (*Ojala, Montoya & Attardi, 1981*; *Boore, 2004*).

**Table 2** **Annotations for the mtgenomes of *G. depressa thoracica* and *G. depressa depressa*.** IGS denotes the length of the intergenic spacer region, for which negative numbers indicate nucleotide overlapping between adjacent genes. H and L denote heavy and light strands, respectively. N.C. indicates non-coding sequence. Gdt and Gdd represent *G. depressa thoracica* and *G. depressa depressa*, respectively.

| Gene | Position | | IGS/bp | | Initiation/Stop Codon | | Anticodon | Coding Strand |
|------|----------|--|--------|--|-----------------------|--|-----------|---------------|
| | Gdt | Gdd | Gdt | Gdd | Gdt | Gdd | | |
| trnI | 1–65 | – | – | – | | | GAT | H |
| trnQ | 65–133 | 6–74 | −1 | – | | | TTG | L |
| trnM | 133–202 | 74–140 | −1 | −1 | | | CAT | H |
| nad2 | 203–1,209 | 141–1,154 | 0 | 0 | ATT/TA | ATT/TAA | | H |
| trnW | 1,210–1,273 | 1,153–1,216 | 0 | −2 | | | TCA | H |
| trnC | 1,266–1,327 | 1,209–1,272 | −8 | −8 | | | GCA | L |
| trnY | 1,328–1,391 | 1,273–1,339 | 0 | 0 | | | GTA | L |
| cox1 | 1,384–2,931 | 1,332–2,879 | −8 | −8 | ATT/TAA | ATT/TAA | | H |
| trnL(UUR) | 2,927–2,991 | 2,875–2,939 | −5 | −5 | | | TAA | H |
| cox2 | 2,992–3,679 | 2,940–3,627 | 0 | 0 | ATT/T | ATT/T | | H |
| trnK | 3,680–3,750 | 3,628–3,697 | 0 | 0 | | | TTT | H |
| trnD | 3,750–3,812 | 3,698–3,761 | −1 | 0 | | | GTC | H |
| atp8 | 3,813–3,968 | 3,762–3,917 | 0 | 0 | ATC/TAA | ATT/TAA | | H |
| atp6 | 3,962–4,636 | 3,911–4,585 | −7 | −7 | ATG/TAA | ATG/TAA | | H |
| cox3 | 4,636–5,419 | 4,585–5,371 | −1 | −1 | ATG/T | ATG/T | | H |
| trnG | 5,420–5,483 | 5,372–5,435 | 0 | 0 | | | TCC | H |
| nad3 | 5,484–5,835 | 5,436–5,787 | 0 | 0 | ATA/T | ATC/T | | H |
| trnA | 5,836–5,902 | 5,788–5,852 | 0 | 0 | | | TGC | H |
| trnR | 5,902–5,964 | 5,852–5,913 | −1 | −1 | | | TCG | H |
| trnN | 5,964–6,027 | 5,911–5,974 | −1 | −3 | | | GTT | H |
| trnS(UCN) | 6,028–6,094 | 5,971–6,034 | 0 | −4 | | | TCT | H |
| trnE | 6,095–6,157 | 6,033–6,095 | 0 | −2 | | | TTC | H |
| trnF | 6,158–6,220 | 6,094–6,159 | 0 | −2 | | | GAA | L |
| nad5 | 6,204–7,925 | 6,140–7,867 | −17 | −20 | ATT/TAA | ATT/TAA | | L |
| trnH | 7,923–7,984 | 7,865–7,928 | −2 | −3 | | | GTG | L |
| nad4 | 7,985–9,317 | 7,929–9,261 | 0 | 0 | ATG/T | ATG/T | | L |
| nad4l | 9,311–9,595 | 9,255–9,539 | −7 | −7 | ATG/TAA | ATG/TAA | | L |
| trnT | 9,599–9,660 | 9,543–9,606 | 3 | 3 | | | TGT | H |
| trnP | 9,661–9,723 | 9,607–9,670 | 0 | 0 | | | TGG | L |
| nad6 | 9,726–10,226 | 9,673–10,143 | 2 | 2 | ATA/TAA | ATT/TAA | | H |
| cob | 10,226–11,365 | 10,143–11,282 | −1 | −1 | ATG/TAG | ATG/TAG | | H |
| trnS(AGN) | 11,364–11,430 | 11,281–11,345 | −2 | −2 | | | TGA | H |
| nad1 | 11,448–12,398 | 11,363–12,313 | 17 | 17 | TTG/TAG | TTG/TAG | | L |
| trnL(CUN) | 12,400–12,464 | 12,315–12,379 | 1 | 1 | | | TAG | L |
| rrnL | 12,465–13,738 | 12,380–13,661 | 0 | 0 | | | | L |
| trnV | 13,739–13,806 | 13,662–13,730 | 0 | 0 | | | TAC | L |
| rrnS | 13,807–14,551 | 13,731–14,281 | 0 | 0 | | | | L |
| A+T-rich region | 14,552–16,109 | – | 0 | – | | | | N.C. |

|  | 5'–atp8 | 7 bp overlap | atp6–3' |  | 3'–nad4l | 7 bp overlap | nad4–5' |
|---|---|---|---|---|---|---|---|
| Gastrolina depressa thoracica | AAATATAACTGAAA | ATGATAA | TAAATTTATTCTCC | | AATTTTTCTTATTT | ATGATAA | GTTTTATTTTTTCT |
| Gastrolina depressa depressa | AATTATAATTGATT | ATGATAA | TAAATTTATTTTCT | | AGTTTTTCAAATTT | ATGATAA | GATTTTTATTTAGT |
| Paleosepharia posticata | AAATATAACTGAAA | ATGATAA | TAAATTTATTTTCA | | ACATTTTCTTCTTT | ATGATAA | AATTTTTATTTGCA |
| Galeruca daurica | AACTATAACTGAAA | ATGATAA | TAAATTTATTTTCT | | ACTTTTTCTTTTTT | ATGATAA | TATATTTATTAAGA |
| Agasicles hygrophila | TCAATTAATTGAAA | ATGATAA | TAAATCTATTCTCA | | TCTTTTTCTTCTTT | ATGATAA | AATTTATTTTCAGA |
| Pyrophorus divergens | AAATTTAACTGAAA | ATGATAA | CAAATCTATTCTCA | | TCTTTTTCTTCTCT | ATGATAA | AGTTTTTGTTTTTT |
| Pyrearimus termitilluminans | TCCATCAATTGAAA | ATGATAA | CAAATCTTTTTTCT | | ACTTTAATATTTT | ATGATAA | AGTTTCTTTTTTTT |
| Damaster mirabilissimus mirabilissimus | ATTCTTAATTGAAA | ATGATAA | CAAATCTTTTTTCA | | TCAATAAATATATT | ATGATAA | AATTTTTATTGATG |

**Figure 2 Sequence alignments of *atp8/atp6* and *nad4/nad4l* of coleopteran insects.** The boxed nucleotides indicate the 7 bp conserved overlaps (ATGATAA).

|  | 3'–nad1 |  | trnSer(UCN) –5' |
|---|---|---|---|
| Gastrolina depressa thoracica | TTTAGTTAACTAAATT | TAGTA | TAAGTCAATAGAAAAT |
| Gastrolina depressa depressa | TTAGTTAATTATTTTT | TAGTA | TAAGTTAATAGAAAAT |
| Paleosepharia posticata | TTTAGTTAATTATTTT | TAGTA | TAAGTTAATAGAAAAT |
| Galeruca daurica | TTTAGTTAATTATTTT | TAGTA | TAAGTTAATAGAGAAT |
| Agasicles hygrophila | TTTAGTTAATTAATTT | TAGTA | TAAGTCAATAGAAAAT |
| Pyrophorus divergens | TTTAGTTAATTAATTT | TAGTA | TAAGTTAATAGGATAG |
| Pyrearimus termitilluminans | TATAGTTAATTAAATT | TAGTA | TAAGTTAATAGGTATC |
| Damaster mirabilissimus mirabilissimus | TATAGTGAATTATTTT | TAGTA | AAAGTTAATAGAGGAG |

**Figure 3 Sequence alignment of the space region between *nad1* and *trnS2* (*UCN*) of coleopteran species.** The boxed nucleotides indicate the 'TAGTA' conserved motif.

**Table 3 Nucleotide compositions in the mitogenomes of *G. depressa depressa*, *G. thoracica depressa*, *Paleosepharia postivata*, *Galeruca daurica* and *Agasicles hygrophila*.**

|  | G. depressa depressa | | | G. depressa thoracica | | | Paleosepharia postivata | | | Galeruca daurica | | | Agasicles hygrophila | | |
|---|---|---|---|---|---|---|---|---|---|---|---|---|---|---|---|
|  | A+T | AT skew | GC skew | A+T | AT skew | GC skew | A+T | AT skew | GC skew | A+T | AT skew | GC skew | A+T | AT skew | GC skew |
| Protein-coding genes | 77.9 | −0.143 | 0.018 | 75.2 | −0.133 | −0.005 | 78.0 | −0.150 | 0.006 | 77.0 | −0.141 | −0.006 | 72.4 | −0.152 | −0.018 |
| First codon position | 73.3 | −0.040 | 0.225 | 71.3 | −0.010 | 0.194 | 72.3 | −0.050 | 0.227 | 72.4 | −0.025 | 0.188 | 68.9 | −0.021 | 0.153 |
| Second codon position | 69.7 | −0.384 | −0.117 | 69.2 | −0.387 | −0.125 | 68.6 | −0.391 | −0.141 | 68.7 | −0.394 | −0.112 | 68.4 | −0.395 | −0.117 |
| Third codon position | 90.6 | −0.040 | −0.135 | 85.2 | −0.03 | −0.153 | 93.2 | −0.049 | −0.211 | 90.1 | −0.042 | −0.210 | 80.0 | −0.058 | −0.128 |
| Protein-coding genes-H strand | 76.4 | −0.087 | −0.106 | 73.3 | −0.050 | −0.162 | 76.6 | −0.112 | −0.110 | 75.1 | −0.089 | −0.134 | 70.7 | −0.084 | −0.145 |
| First codon position | 70.8 | 0.057 | 0.132 | 68.4 | 0.112 | 0.079 | 69.7 | 0.032 | 0.140 | 69.5 | 0.064 | 0.098 | 66.7 | 0.079 | 0.071 |
| Second codon position | 68.0 | −0.361 | −0.221 | 66.8 | −0.364 | −0.209 | 67.2 | −0.361 | −0.217 | 67.2 | −0.368 | −0.202 | 67.3 | −0.70 | −0.203 |
| Third codon position | 90.2 | 0.005 | −0.470 | 84.8 | 0.066 | −0.562 | 92.8 | −0.041 | −0.681 | 88.5 | 0.001 | −0.557 | 78.1 | 0.025 | −0.388 |
| Protein-coding genes-L strand | 80.1 | −0.227 | 0.246 | 78.8 | −0.246 | 0.286 | 80.4 | −0.206 | 0.229 | 80.2 | −0.219 | 0.255 | 75.3 | −0.256 | 0.224 |
| First codon position | 77.0 | −0.167 | 0.410 | 76.1 | −0.149 | 0.417 | 76.5 | −0.170 | 0.405 | 77.0 | −0.153 | 0.380 | 72.5 | −0.168 | 0.311 |
| Second codon position | 72.1 | −0.431 | 0.053 | 71.6 | −0.433 | 0.044 | 70.8 | −0.437 | −0.005 | 71.1 | −0.443 | 0.051 | 70.1 | −0.434 | 0.035 |
| Third codon position | 91.2 | −0.116 | 0.429 | 88.7 | −0.177 | 0.617 | 93.8 | −0.062 | 0.659 | 92.6 | −0.110 | 0.657 | 83.1 | −0.183 | 0.414 |

The A+T content, AT-skew and GC-skew of *G. depressa thoracica*, *G. depressa depressa* and three other Chrysomelidae were calculated, respectively. All PCGs have a high A+T percentage (70%) (Table 3). This indicated that PCGs have the high background mutational pressure toward AT nucleotides at the third codon position ((*Kim et al., 2014*).

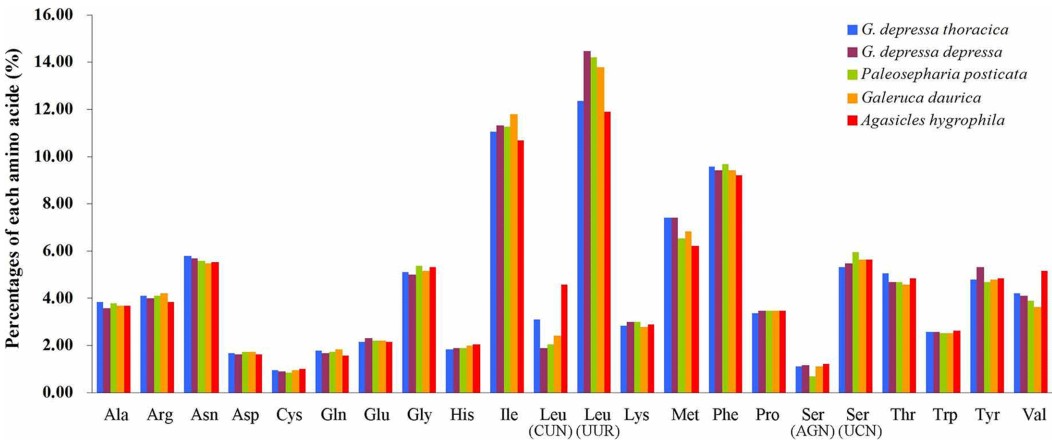

**Figure 4** **Percentages of amino acid usage in mitochondrial proteins of five species.** Each amino acid is represented by the three-letter abbreviation. Note that leucine and serine are each coded by two different genetic codons, and listed separately.

The relative synonymous codon usage (RSCU) showed that the most frequently used amino acids in these two mitogenomes were Leu, Ile, Phe and Met. TTA for Leu, ATT for Ile, TTT for Phe and ATA for Met were the most popular codons in *G. depressa thoracica* and *G. depressa depressa* mitogenomes (Figs. 4, 5). The sum of the most popular amino acids, like Leu, Ile, Phe and Met, varied from 38% (*Agasicles hygrophila*) to 42.23% (*G. depressa depressa*). All PCGs are rich with A or T nucleotides, which was also found in other coleopteran insects (*Kim et al., 2009*; *Du et al., 2016*).

The codon families in PCGs were encoded by H and L strands (Fig. 6), respectively. The amino acid abundance of PCGs on H and L strands have different skewness, which led to an amino acid usage unbalance in PCGs. PCGs on the H-strand were more TA-skewed than CG-skewed, whereas the PCGs on the L-strand had a higher frequency of T and G, in which the first and second codons were skewed toward A nucleotide (*Pons et al., 2010*). Similarly, the third codon was also A/T biased (Fig. 6). Overall, this nucleotide preference lead to 13 PCGs having a higher percentage of Leu and a lower percentage of Cys. However, *nad* genes on the L-strand (*nad1*, *nad5*, *nad4* and *nad4L*) encode more Cys (≥2) than any other genes on the H-strand (only 1–2) (Table S3). Four-fold degenerate codon usage was obviously A/T biased in the third position, and two-fold degenerate codon usage showed a similarly biased pattern, with A/T favored over G/C in the third position (Fig. 6). Both patterns were in agreement with the AT-biased content exhibited by PCGs. The most obvious bias of amino acid usage was due to structural/functional requirements of PCGs, which is well represented by the distribution of the Cys codon family. For example, mitochondrial *nad* genes have the highest Cys content, which are essential structures and help to form intra- and inter-chain disulfide (*Mishmar et al., 2006*).

### Transfer RNA genes

Twenty-two tRNAs were found in *G. depressa thoracica* and ranged from 62 bp to 71 bp in length with 14 tRNAs on the H-strand and eight on the L-strand. Except for *trnSer* (*AGN*),

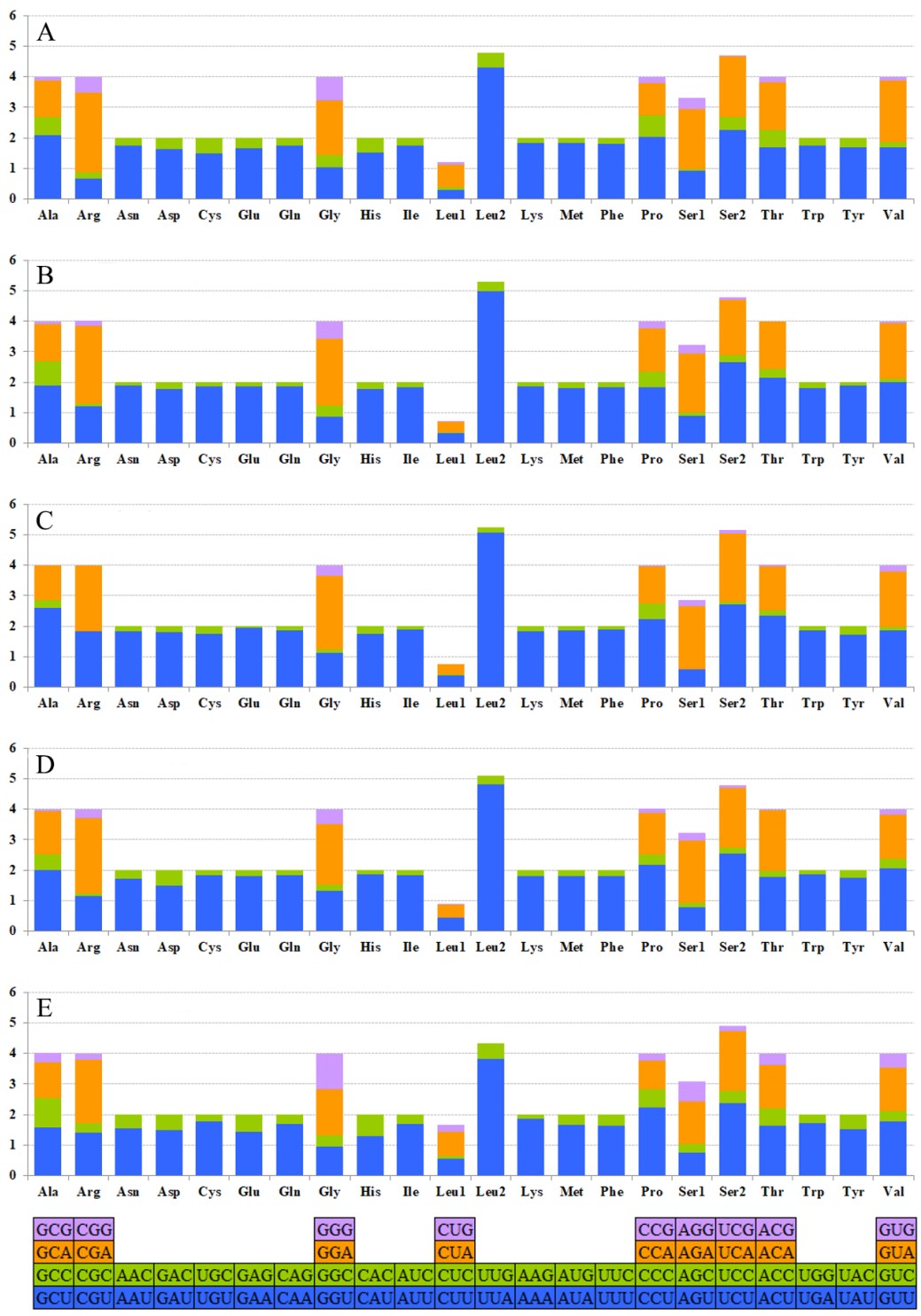

**Figure 5** **The mitogenome relative synonymous codon usage (RSCU) across five coleopteran insects.**
(A) *Gastrolina depressa thoracica*. (B) *G. depressa depressa*. (C) *Paleoseparia posticata*. (D) *Galeruca daurica*.
(E) *Agasicles hygrophila*. Codon families are provided on the *x*-axis.

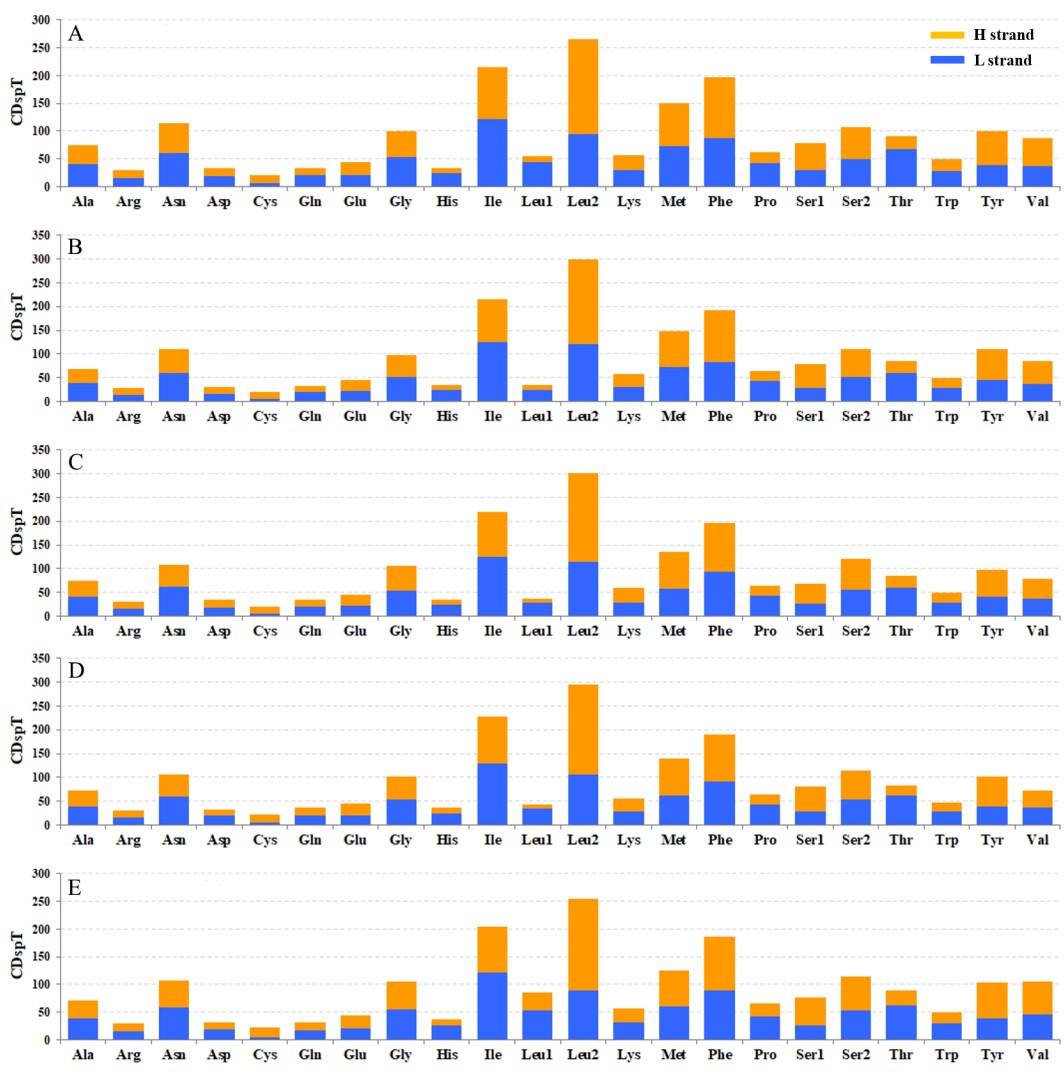

**Figure 6** **Codon distribution in five coleopteran insects. CDspT, codons per thousand codons.** (A) *Gastrolina depressa thoracica*. (B) *G. depressa depressa*. (C) *Paleoseparia posticata*. (D) *Galeruca daurica*. (E) *Agasicles hygrophila*. Codon families are provided on the *x*-axis. Within each family, the percentage of codons located on the H strand or L strand is colored with blue or orange, respectively.

all tRNAs were folded into the typical clover-leaf structure (Fig. S2), containing an amino acid acceptor arm (7 bp), T Ψ C arm (3–5 bp), dihydorouridine (DHU) arm (3–4 bp), anticodon arm (3–5 bp) and a variable extra arm. In the mitogenome of *G. depressa depressa*, 21 tRNAs were identified (Fig. S2) and the length of these genes were between 62 bp to 70 bp, in which 13 and 8 tRNAs were encoded by the H-strand and L-strand, respectively. Twenty tRNAs had the typical clover-leaf structures, containing an amino acid acceptor arm (7 bp), T Ψ C arm (3–5 bp), DHU arm (3–5 bp), anticodon arm (3–5 bp) and a variable extra arm. However, *trnSer* (*AGN*) did not have the DHU arm, and formed a simple-loop structure, which is a common phenomenon in many insect mitogenomes (*Wolstenholme, 1992*).

In tRNAs, the typical DHU arm is encoded by 3–4 nucleotides; the AC arm and the T ΨC arm vary from 3–5 bp; and the variable loops range from 4–6 bp. In general, the size of the variable loop and the D-loop affects the length of the tRNA (*Navajas et al., 2002*). In the comparison of both mitochondrial genomes (Fig. S2), the major base pairs, A-T and G-C in stem regions are Watson-Crick pairs, while G-U wobble and mismatched pairs can also be observed. A total of 18 and 12 G-U wobbles were observed in the mitogenomes of *G. depressa thoracica* and *G. depressa depressa*, with 10, four, three, one and three, four, four, one base pairs in AA stem, D stem, AC stem and T stem of the tRNAs, respectively. In these wobbles, three were in the D stems of *trnGln*, *trnPro* and *trnHis*, and one was in the AA stem of *trnCys*. Except for *trnIle*, three (one AG in *trnTrp*, and two UU in *trnLeu* (*UUR*) and *trnLeu* (*CUN*)) and two (one AG in *trnTrp* and one UU in *trnLeu* (*UUR*)) mismatched base pairs were found in the tRNAs of *G. depressa thoracica* and *G. depressa depressa* mitogenomes, respectively. Among the six mismatched base pairs, four were located in the amino acid acceptor arms, and one UU pair was located in the anticodon arm of *trnLeu* (*CUN*). These mismatches, which are apparently deleterious, can be repaired and corrected by a putative RNA editing process (*Masta & Boore, 2004*). From protozoa to plants and metazoan animals, tRNA editing events have been observed in a variety of mitochondrial tRNAs (*Laforest, Roewer & Lang, 1997*; *Alfonzo et al., 1999*; *Leigh & Lang, 2004*; *Grewe et al., 2009*; *Knoop, 2011*).

The percentage of identical nucleotides (%INUC) for both insects was calculated (Table S4). *TrnD*, *trnN*, *trnR*, *trnE*, *trnW*, *trnS* (*UCN*) and *trnP* showed a higher %INUC (>90%) in both insects, and the first six were located on the H strand (Table 2). Only *trnF* and *trnL* (*CUN*) on the L-strand had a lower %INUC (<80%), indicating the level of conservation was positively H strand-biased.

## Phylogenetic analyses

Phylogenetic trees were constructed based on 13 PCGs nucleotide sequences from 36 species (Fig. 7A) and their corresponding amino acid sequences (Fig. 7B) by maximum likelihood (ML) and Bayesian Inference (BI) analyses. Overall, BI analyses provided more resolution with strong supports. *G. depressa thoracica* and *G. depressa depressa* formed one clade (BPP/BS = 1/100), and had a closer relationship with species from Galerucinae than *G. intermedia* from Chrysomelinae.

The family Chrysomelidae was divided into three major clades in both nucleotide and amino acid sequence trees using ML and BI analyses (Fig. 7). Chrysomelinae and Galerucinae consistently formed a clade, and the other four subfamilies, Bruchinae, Criocerinae, Donaciinae, and Spilopyrinae, were clustered in a clade. The phylogenetic analyses in this study were also well supported by previous analyses (*Farrell, 1998*; *Reid, 2000*; *Farrell & Sequeira, 2004*; *Gómez-Zurita et al., 2007*; *Bocak et al., 2014*; *Song et al., 2017*), in which the clade of Eumolpinae + Cryptocephalinae + Cassidinae were close to Galerucinae and Chrysomelinae, with the latter observed in all the phylogenetic trees.

In the phylogenetic analyses of amino acid sequences, Orsodacninae had a closer relationship with the clade of Lepturinae + Necydalinae in the superfamily Cerambycidae. This implied that Orsodacninae might be evaluated outside Chrysomelidae (*Haddad*

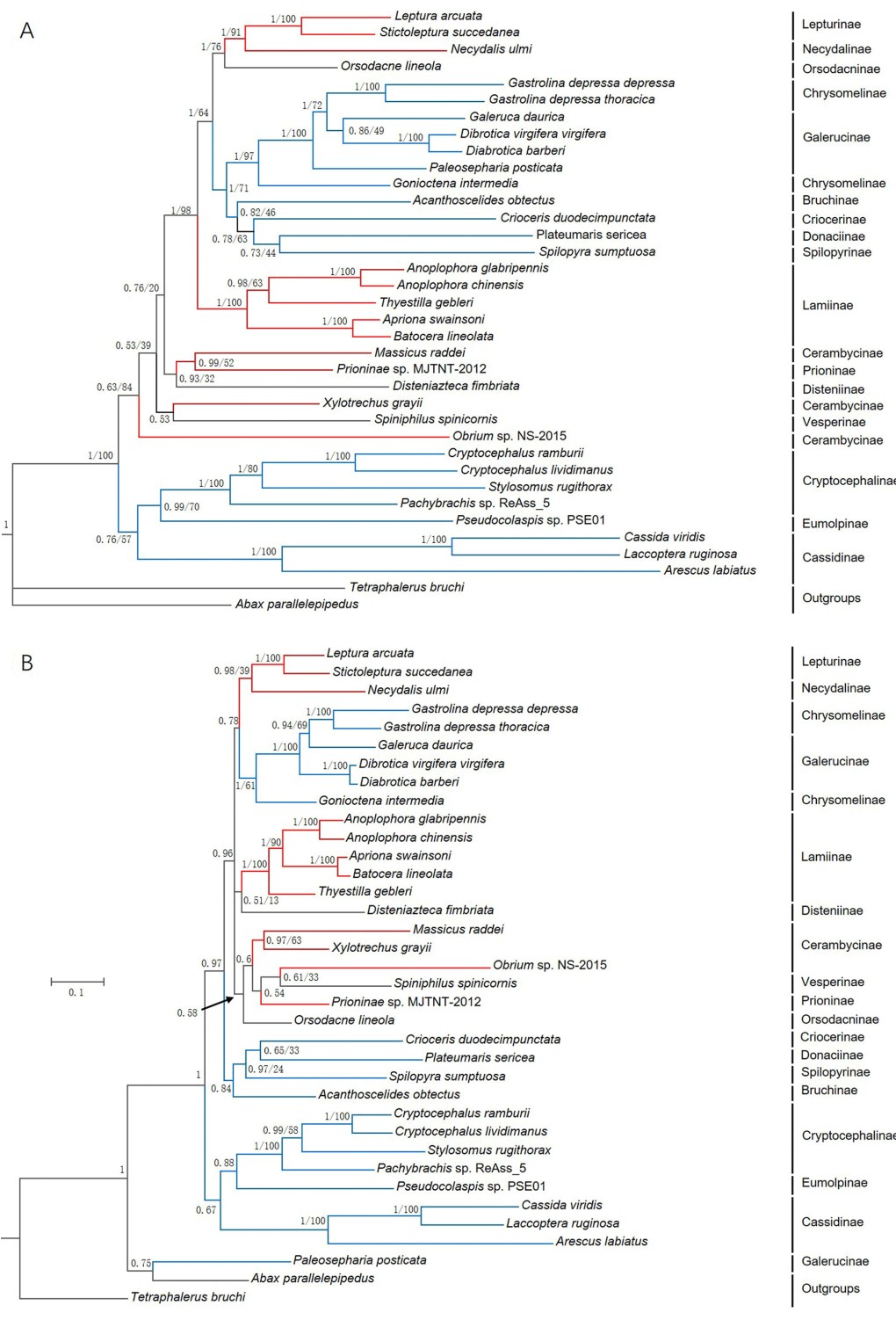

**Figure 7** **Phylogenetic trees based on (A) the nucleotide and (B) the amino acid datasets for 13 protein-coding genes from the mitochondrial genomes of 36 species.** All the probability values and bootstrap values of the branches were indicated, except for those where the topology of ML and BI was different from different datasets.

*& Mckenna, 2016*). *Reid (1995)* was also against placing Orsodacninae within the Cerambycidae based on mitochondrial nucleotide sequences analyses. The representative species from Lamiinae formed a clade in the four phylogenetic trees, while the interrelationships within Cerambycinae remained unclarified due to insufficient mitogenome and genome data from these insects. Certainly, as more mitogenomes and genomes of insect species are available in databases, phylogenetic analyses will be more reliable and convincing.

## CONCLUSIONS

In this study, the complete mitogenome of *G. depressa thoracica* (16,109 bp) and the partial mitogenome of *G. depressa depressa* (14,277 bp) were sequenced and analyzed. In the mitogenomes of *G. depressa thoracica* and *G. depressa depressa*, the overall mitochondrial gene order and orientation were identical with some exceptions. The motifs, 'TAGTA' between *trnSer* (*UCN*) and *nad2*, and 'ATGATAA' between *atp8* and *atp6*, were found to be more conserved than 'ATGATAA' between *nad4* and *nad4L* in the mitogenomes of Polyphaga, which was 'ATGTTAA' in the Adephaga, Myxophaga and Archostemata. Phylogenetic analyses showed that *G. depressa thoracica* and *G. depressa depressa* consistently formed a clade. Within Chrysomeloidea, the sister relationships of ((Eumolpinae + Cryptocephalinae) + Cassidinae), and the close affinity of Galerucinae + Chrysomelinae were confirmed with high nodal supports. We believe that the mitogenomes of *G. depressa thoracica* and *G. depressa depressa* will be useful for further studies of molecular classification, and coleopteran mitogenome architecture and phylogenetics.

### Funding

This research was supported by the National Natural Science Foundation of China (No. 31770692) and the key research and development project of Shaanxi Province (2017NY-105). The funders had no role in study design, data collection and analysis, decision to publish, or preparation of the manuscript.

### Grant Disclosures

The following grant information was disclosed by the authors:
National Natural Science Foundation of China: No. 31770692.
The key research and development project of Shaanxi Province: 2017NY-105.

### Competing Interests

The authors declare there are no competing interests.

### Author Contributions

- Qiqi Wang conceived and designed the experiments, performed the experiments, analyzed the data, contributed reagents/materials/analysis tools, prepared figures and/or tables, authored or reviewed drafts of the paper, approved the final draft.

- Guanghui Tang conceived and designed the experiments, contributed reagents/materials/analysis tools, authored or reviewed drafts of the paper, approved the final draft.

## DNA Deposition

The following information was supplied regarding the deposition of DNA sequences:

The mitochondrial sequences of *Gastrolina depressa depressa* and *G. depressa thoracica* are accessible via GenBank accession numbers MF198407 and MF198406, respectively.

## Supplemental Information

Supplemental information for this article can be found online at http://dx.doi.org/10.7717/peerj.4919#supplemental-information.

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
