# Peer review of "The mitochondrial genomes of two walnut pests, Gastrolina depressa depressa and G. depressa thoracica (Coleoptera: Chrysomelidae), and phylogenetic analyses"

_PeerJ, doi:10.7717/peerj.4919_

## Round 0.1 · original submission · Major Revisions

The manuscript, although interesting, requires further work before publication. In particular, the authors are directed to the comments of reviewer 3 that should be carefully considered and fully addressed in the revision. In particular, although the rationale for the study stated in the Introduction was to resolve the taxonomic status of the 2 insects, there was ultimately no resolution proposed and the discussion focused almost exclusively on issues with in the much larger superfamily. All reviewers suggest revisions to the data presentation in the figures. In addition, a number of relevant recent references are suggested for inclusion to enhance the interpretation of the data. The manuscript also needs substantial clarification of English grammar and should be proofread by a native English speaker. Two reviewers have submitted annotated manuscripts that illustrate small points that need revision and the third reviewer has provided detailed comments and suggestions.

·

Basic reporting

I thought the paper is meet the Peerj published standards. The English is clear and can be understand. Two mt genomes of Coleoptera were sequenced and compared.
But the paper need some revisions.
I thought the all sequenced download from GenBan should be referenced. So the authors should added the references of 36 mt genomes of Coleoptera from GenBank. The other literature references is no problem.
The figs 6 and 7 should be added all bootstrap and BPP number above the branch not only one value.

Experimental design

The experimental desing and analysis method is no problem.
If the authors can obtain the complete mt genomes of Gastrolina depressa depressa, the paper can deeply discuss the difference within two subspecies. I just find one pair of primers you designed in Table 1. If you failed to sequence the AT region you can sequence this region by colon method. I also suggest the authors compare the genetic divergence in all genes and whole genomes, which can give us the more different informations between two subspecies.

Validity of the findings

Two mt genomes were sequenced.
Some different gene charachters were found in two mt genomes.
Conclusion is well.

Additional comments

Two mt genomes of Coleoptera were sequenced and compared in the paper. The paper is meet the Peerj published standards. The English is clear and can be understand.
1. But the paper need some revisions. I thought the all sequenced download from GenBan should be referenced. So the authors should added the references of 36 mt genomes of Coleoptera from GenBank. The other literature references is no problem. The figs 6 and 7 should be added all bootstrap and BPP number above the branch not only one value.
2. If the authors can obtain the complete mt genomes of Gastrolina depressa depressa, the paper can deeply discuss the difference within two subspecies. I just find one pair of primers you designed in Table. If you failed to sequence the AT region you can sequence this region by colon method or design the other primers. I also suggest the authors compare the genetic divergence in all genes and whole genomes, which can give us the more different informations between two subspecies.

Other minor revsions is revised in PDF. Please see the PDF.

Reviewer 2 ·

Basic reporting

The English writing is generally professional, only with some minor errors in a few word ore sentence expression. All the errors was listed or labelled in the revised manuscript with detail and the authors was suggested to checked them carefully and correct them.

Experimental design

no comment.

Validity of the findings

no comment.

Additional comments

1. The studies firstly reported the mitochondrial genomes of two walnut pests,Gastrolina depressa depressa and G. depressa thoracica (Coleoptera: Chrysomelidae), and conducted the relevant phylogeny within Chrysomeloidea. This report is valuable for enriching the mitogenomic data of beetles and related future colepteran phylogenetic studies.
2. Some minor errors of word and sentence expressions in the English writing were all listed or labelled in the revised manuscript, and the authors was suggested to check and correct them carefully.
3. The two phylogenetic trees shown in the Figure 6 and Figure 7 of the manuscirpt were suggested to be incoporated into one consesus tree and shown in one figure.

Annotated reviews are not available for download in order to protect the identity of reviewers who chose to remain anonymous.

Reviewer 3 ·

Basic reporting

The mitochondrial genomes of two walnut pests, Gastrolina depressa depressa and G. depressa thoracica (Coleoptera: Chrysomelidae), and phylogenetics within Chrysomeloidea (#21999) by Qiqi Wang, and Guanghui Tang

The authors describe the mitochondrial genomes of two walnut leaf pests, Gastrolina depressa depressa and Gastrolina depressa thoracicia, illustrating similarities in gene content and gene arrangement across these two insects, as well as conserved elements found in other coleopteran taxa. In addition, they use nucleotide and amino acid sequences from the 13 protein-encoding genes from these two taxa, along with 34 other species within the Chrysomeloidea (and outgroups), to infer the pattern of relationships within this large superfamily of beetles using ML and Bayesian analyses. Their phylogenetic analyses provided consistent support for the monophyly of Eumolpinae + Cryptocephalinae + Cassidinae, but little resolution for basal relationships within the Chrysomeloidea.

This study is a little unusual in that two central taxa at the focus of the study are presented as subspecies of a single species Gastrolina depressa. If the authors were interested in generating two new mtDNA genomes, in part to help better understand mt genome dynamics and phylogenetic relationships within the Chrysomeloidea, why choose two subspecies? This fact alone made the central rationale for this study seem a little tenuous. Despite their subspecies designation, however, the authors present these two taxa as if they were separate species, noting differences in morphology, and referring to past work on their systematics, occurrence and spatial distribution. Yet they never comment on the species vs. subspecies status specifically. This is odd, particularly when Ge et al. 2003 indicate that these two subspecies should be considered two separate species: Gastrolina depressa and Gastrolina thoracica. A discussion of this paper would seem to be important for this study. Why was this recommendation not followed?

In addition, the authors also claim on lines 71/72 that this study was undertaken to understand the taxonomic status of these two insects, and the phylogenetic relationships within the “Chrysomeloidae” (which I assume is a typo – it should be Chrysomeloidea or Chrysomelidae). Yet, in the Results/Discussion, the authors never address how their data have any bearing on the taxonomic status of Gastrolina depressa depressa and Gastrolina depressa thoracica, other than that they group together in their phylogenetic analysis. Hence, this study appears to be somewhat muddled in terms of its focus, and the choice of the focal species used. A stronger introduction to the species and their current taxonomic status would have helped to resolve some of these issues.

The conclusions that can be drawn from their phylogenetic analyses based on the 13 protein-encoding genes of 36 taxa are also somewhat limited given the superficial coverage given to their results and discussion. There also appear to be some important outcomes of their analyses that are not discussed at all, but need to be. For instance, they state on line 209-211 that Galerucinae + Chrysomelinae were monophyletic (albeit only in the tree derived from the nucleotide dataset), but don’t mention that the difference in analyses is because Paleosepharia falls outside the Chrysomeloidea clade entirely and groups with the outgroup taxon in the amino acid dataset. Surely some explanation is warranted here. Likewise, there is a focus on the monophyly of Galerucinae + Chrysomelinae (despite the weird results of the amino acid dataset) but no mention of the fact that neither analysis shows either subfamily as monophyletic. Despite this, latter on in the Conclusions (line 224-226), Galerucinae + Chrysomelinae are referred to as having high nodal support as sister taxa.

There are other issues too, including recent (2016, 2017) references that would appear to be relevant to this study that were not cited in this paper, including:
• Ming-Long Yuan, Qi-Lin Zhang, Li Zhang, Zhong-Long Guo, Yong-Jian Liu, Yu-Ying Shen, Renfu Shao. 2016. High-level phylogeny of the Coleoptera inferred with mitochondrial genome sequences. Molecular Phylogenetics and Evolution 104: 99-111, ISSN 1055-7903, https://doi.org/10.1016/j.ympev.2016.08.002.
• Hongli Zhang, Nian Liu, Zhiping Han, Jianxia Liu. 2016. Phylogenetic analyses and evolutionary timescale of Coleoptera based on mitochondrial sequence. Biochemical Systematics and Ecology 66: 229-238. ISSN 0305-1978, https://doi.org/10.1016/j.bse.2016.04.014.
• Haddad, S. and D. McKenna. 2016. Phylogeny and evolution of the superfamily Chrysomeloidea (Coleoptera: Cucujiformia). Systematic Entomology 41: 697–716 DOI: 10.1111/syen.12179
• Nan Song, Xinming Yin, Xincheng Zhao, Junhua Chen & Jian Yin. 2017. Reconstruction of mitogenomes by NGS and phylogenetic implications for leaf beetles. Mitochondrial DNA Part A. [Epub ahead of print].
The most recent of these may not have been seen by the authors because it is a 2017 publication, only currently available as an epub. Nevertheless, it is clearly overlapping with the present study in terms of focus on mtDNA genomes and chrysomelids.

In addition, this study is also undermined by grammatical problems and spelling mistakes, which can lead to challenges in interpretation in several places. Although this is understandable if the authors’ first language is not English, these errors must be corrected before publication.

In terms of figures:
Figure 1: The region that was not sequenced should be clearly noted for G. depressa depressa. It was also unclear to me why the genome size could not be estimated for this species based on the size of amplification products. If this was a sequencing issue (lines 136 – 137), then amplification products could be used to get an estimate of the total genome length.

Figures 2 and 3 could be included as supplementary information.

Figures 6 and 7. It is unclear where the phylogram came from that was used to map the BP and PP values. Is this from the ML or Bayesian analysis? Why are different rules used for the two trees in terms of BBP and PP reporting? For instance, PP>60 and BBP>0.7 are reported in Figure 6, and PP>75 and BBP>0.6 reported in Figure 7. What are the standard metrics for reporting PP and BBPs?

Experimental design

As mentioned above, the central rationale for this study is unclear, in part because the authors undertook a detailed study of the mtDNA genomes of two closely related taxa in one genus (Gastrolina), and then attempted to use these genomes to better understand phylogenetic relationships within the diverse superfamily Chrysomeloidea. In doing so, they have given short shrift to the (apparent) initial focus of this study – the taxonomic status of of two walnut leaf pests, Gastrolina depressa depressa and Gastrolina depressa thoracicia – and an extremely superficial overview of their analysis/results pertaining to phylogenetic relationships within the family. Similarities/differences between the analyses based on nucleotide vs amino acid datasets are poorly summarized, and glossed over very quickly. Odd results such as that pertaining to the placement of Paleosepharia in the amino acid analysis are not addressed.

On lines 92-93, the authors report that the PCR products were 1.1 to 1.4kb in length, but then provide a list of primer pairs in Table 1 that are associated with amplification products ranging in size from 1.1 – 4.1 kb. How can this be so?

It would have been interesting to know too, why the authors didn’t elect to try longer amplifications using LA-PCR protocols, and attempt to size the mtDNA genome of Gastrolina depressa depressa even though they couldn’t sequence through the AT-rich region.

Giving the size of the partial genome in the Abstract (for comparison to G.depressa thoracica) isn't very meaningful without some explanation as to why this was not completely sequenced, and that the sequenced region included all the standard 37 genes.

The presentation of a Results/Discussion section as a combined unit rather than as separate entities obscures the fact somewhat that this study really lacks a comprehensive discussion of the results, including an attempt to interpret the relevance of the results to the taxonomic status of the two focal taxa (an objective of the study), or the higher level phylogenetic relationships within the Chrysomeloidea (a second objective).

Validity of the findings

The presentation and reporting of the two mitochondrial genomes appears to have been done well, with the standard features reported and unique and conserved elements of these genomes highlighted, with a limited comparison to some other coleopteran mtgenomes. The authors purport to have provided a detailed comparative investigation into the composition of mitogenomes for Chrysomelinae (line 70-71). I’m not sure that I saw this here, but certainly some comparisons were made to other taxa.

Bigger concerns associated with this study relate to the phylogenetic analyses presented in Figure 6 & 7, which are reported and discussed somewhat superficially, with minimal attempts to discuss strange outcomes, and with errors in reporting mentioned above. The statement on line 205 – “Regardless sequencing or database errors in some taxa” also lead one to question the quality of some of the data included in this study, and really requires the authors to qualify what they mean by this statement, and to ensure that they winnowed out data that were suspect prior to undertaking their analyses.

Also, frequent issues with typographical and grammatical errors lead to difficulties in understanding what the authors are trying to say in places, as well as in interpreting the data. These should be resolvable, however, with editorial assistance.

Additional comments

I felt that this paper would be a stronger contribution, as written, without the phylogenetic component. This seems to have distracted the authors from providing more of a focus on the mitochondrial genomes of two focal species, particularly with respect to providing a more comprehensive comparison of the structure and composition of mtDNA genomes across the Chrysomeloidea. If the authors want to include the phylogenetic component, then I think that it is important to interpret their results more fully, explore the odd placement of taxa (as noted above), and incorporate more of the recent literature into their discussion.

---

## Round 0.2 · Major Revisions

Reviewer 3 still has 2 major concerns about the manuscript. I agree that both need to be addressed before the manuscript can be finally accepted.

1. English spelling and grammar needs to be improved so that readers can clearly understand the manuscript. I agree with the reviewer that the manuscript should be carefully checked and edited by a native English speaker.

2. The reviewer points out that the stated aim of the study in the Introduction "to understand the taxonomic status of these two insect taxa" receives little attention or resolution in the Discussion. The reviewer suggests that perhaps a solution to this is to alter the Introduction statements about the aim to understand the taxonomic status of the two taxa, and instead center the study around understanding where the two taxa fit within the Chrysomeloidea. I suggest that it could be a good solution so consider revising the paper with this idea in mind. As an alternate, you could consider changing the Introduction to propose that the study had 2 aims - one to understand the taxonomic status of the 2 taxa, and the other to understand the where these taxa fit within he Chrysomeloidea. Then in the Discussion, you could make clear statements about whether the data gathered resulted in clear answers to aim 1 or aim 2 or both.

·

Basic reporting

One minor suggestion is that those different topology of nodals in Figure 6 can be used as 0.54/- ..... not just 0.54, which maybe give reader clear to understand.

Experimental design

The experimental designing and analysis method is no problem.

Validity of the findings

The data is robust and conclusion are well stated.

Additional comments

I thought the paper is meet the Peerj published standards now. It can be accepted.

Reviewer 3 ·

Basic reporting

I appreciate the fact that the authors have put considerable effort into responding to criticisms of the reviewers, but I still find the manuscript unacceptable for publication in its current state.

The writing still needs to be improved substantially. There are sentences, even in the Abstract, that are difficult to interpret. I am sympathetic to the challenges the authors are facing here, but it is in the authors’ best interests to ensure that their writing is clear and understandable. I would encourage the authors to seek editorial support from a native English speaker - some of the same problems in version 1 of the manuscript remain in this version. Spelling mistakes also abound - e.g. Celeoptera (L. 42)and Chrysommeloidea (L49) - which suggest a lack of attention to detail.

The authors continue to state that their results will help to understand the taxonomic status of these two insect taxa, but then once they have their results, shy away from trying to comment on this other than to say that their data are insufficient to do so. This still is problematic for me. Perhaps a solution to this is to avoid making this statement entirely, and place this study firmly within the context of understanding where their two taxa fit within the Chrysomeloidea.

Experimental design

In my last review, I criticized the paper because the central rationale for this study seemed a little tenuous – because the authors claimed to be using mtDNA genomes of two subspecies to “help to examine the detailed classification status of both insects” but yet never addressed what their results had to say about the relationship of these two taxa. This seemed odd, particularly when Ge et al. 2003 indicated that these two subspecies should be considered two separate species: Gastrolina depressa and Gastrolina thoracica. In the current revision the authors do address this. They state that “the existing molecular data especially mitogenomes of species in Chrysomelinae, are not sufficient enough to detailedly classified the two newly sequenced insects” or variations thereof throughout the manuscript. I am not sure what they mean by this or exactly what data they feel that they would need to make this sort of determination. Barcoding approaches using COI gene regions are routinely used as coarse metrics of species-level status within animal taxa, as so could be referred to if nothing else, and presumably there are lots of associated studies to compare their genetic divergences to. If they don’t feel that they have enough data to lend their support to species or subspecies status, what data do they need?

In the abstract the authors draw attention to several results, two of which I am puzzled by. They give a fair amount of space to the presentation of repetitive elements in three regions of the genome, but fail to demonstrate why this is important or significant. Secondly, they draw attention to the fact that G. depressa thoracica and G. depressa depressa were found to be strongly supported as sister taxa in their analyses. This seems hardly a surprise unless the classification of these two species/subspecies was really off. These were the only two members of the genus examined in their phylogenetic analysis, and two of only three species in the subfamily Chrysomelinae included in their analyses. Hence, I wonder if there aren’t other aspects of their results which are worth highlighting in the Abstract instead of these two points above.

Validity of the findings

The authors have done a lot of work to generate and annotate the mtDNA genomes of their two focal taxa, and to undertake the phylogenetic analyses to place these taxa within a broader phylogenetic setting. My major challenge as a reviewer of this manuscript is with the context of this study. I don’t think that there is a good match with what the study proposes to do, and what it actually does. As explained above, this needs to be addressed by either changing the overarching goals, or trying to examine the data so that some commentary can be made towards satisfying those objectives. A second issue is with the writing. As good as the writing is considering that the authors are (I assume) not native English speakers, improvements are necessary to ensure that the meaning of the authors is fully understood by the reader. I think that this can be resolved by having the manuscript read/revised by a native English speaker.

---

## Round 0.3 · Minor Revisions

Dear Drs. Wang and Tang.

Thank you for the revisions to your manuscript. In terms of the science, I think that your revisions are strong and appropriate. However, I find that there are still substantial problems with correct English language. I have quickly read through your manuscript and I have provided suggested changes in wording in multiple places on the attached PDF.

1. Yellow highlights show words or phrases that are in better English. Please check these to be sure that I have not misunderstood what you were saying. If the yellow changes are acceptable, please transfer them to the Word version of your document and then resubmit.

2. Blue highlights show sentences that I could not fully understand and therefore I was not able to suggest corrections. Please revise these sentences and have them re-checked by an English speaker for correctness before resubmission of your Word manuscript.

Best regards.

---

## Round 0.4 · accepted · Accept

Thank you for completing the revisions and congratulations on the acceptance of the manuscript.

Ken Storey

#